# Comment on Ssuna et al. Animal Welfare Guidelines for International Development Organisations in the Global South. *Animals* 2024, *14*, 2012

**DOI:** 10.3390/ani14233446

**Published:** 2024-11-28

**Authors:** Jo Hockenhull, Faith A. Burden, Cara Clancy, Tamlin Watson, Laura M. Kubasiewicz

**Affiliations:** The Donkey Sanctuary, Slade House Farm, Weston, Sidmouth EX10 0NU, UK; faith.burden@thedonkeysanctuary.org.uk (F.A.B.); cara.clancy@thedonkeysanctuary.org.uk (C.C.); tamlin.watson@thedonkeysanctuary.org.uk (T.W.); laura.kubasiewicz@thedonkeysanctuary.org.uk (L.M.K.)

International Development Organisations (IDOs) undertake critical work to improve the lives of people across the world, particularly in the Global South. Historically, however, they rarely consider animal welfare, a significant oversight given the value of animals for providing personal and/or commercial transport, traction and production for those in low- and lower-middle-income countries. With awareness growing that humans, animals and the environment are part of an interconnected system, a more holistic One Health One Welfare approach needs to be taken whereby the needs and wellbeing of all elements are considered rather than taking a solely anthropocentric stance. With this in mind, we were pleased to see the recent publication by Ssuna et al. [1] reporting animal welfare guidelines for IDOs working in the Global South.

As the authors highlight, the activities of IDOs can impact animals directly, for example, through the provision of livestock to individuals or communities via in-country projects, or indirectly through their strategic advice to governments. Ssuna et al. developed their guidelines for IDOs to promote animal welfare in IDO projects using a process of stakeholder engagement, with the final guidelines ultimately gaining consensus support. 

Yet there seems to be a significant oversight in the guidelines produced: the omission of reference to the needs and welfare of working animals, despite the crucial role they play in human lives and livelihoods in the Global South. By “working animal” we mean any animal that is kept by humans to perform tasks. This could include therapy animals; animals who are used in policing and military activity; animals used in sports, leisure and tourism, e.g., circus animals; and animals performing manual labour either for subsistence or income generation. Of particular relevance to the Global South are animals who are used for traction and transport, such as draft and pack horses, mules, donkeys, oxen, buffalo and camelids. Working equids, for instance, support the livelihoods of some of the most marginalised communities in the world, and working animals, in general, support the livelihoods of between 300 and 600 million people globally [2]. Working animals also play an intrinsic part in some IDO activities, for example, being used to deliver humanitarian aid to remote communities that cannot be reached by vehicles [3]; these situations require animals to work in extreme conditions, making guidelines to support their welfare of particular importance. It would be interesting to know if working animals were not specifically mentioned in the guidelines because they were simply not mentioned by the stakeholders who engaged in their development or were not observed in the projects visited. Or did the authors make a decision not to explicitly include them in the guidelines? And if this was the case, we would be interested in why. 

Working animals are frequently overlooked in national and international policy and strategy [4,5] and given the above, it would have been good to see them included in these guidelines. The authors state that their guidelines are intended to complement the WOAH Terrestrial Animal Health Code but, whilst the code does contain some guidance for maintaining the welfare of working equids (Chapter 7.12) [6], these are non-specific to IDO activities, and the welfare of other working animals such as oxen, buffalos and camelids is entirely overlooked. The current guidelines would have been the ideal place to address this gap in guidance.

Those advocating for consideration of working animal welfare have fought long and hard for their inclusion in strategies, policies and guidelines aimed at improving the lives of animals worldwide. The authors of the current guidelines have a unique opportunity to ensure working animals are included on the agenda, and we would strongly support an update to the guidelines to include the specific needs of animals in their working environment, thereby assisting efforts to finally dispel their invisible worker status [4].

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
