# Peer review of "Comment on Ssuna et al. Animal Welfare Guidelines for International Development Organisations in the Global South. Animals 2024, 14, 2012"

_animals, 2024, doi:10.3390/ani14233446_

Round 1

Reviewer 1 Report

Comments and Suggestions for Authors

The submitted comment states:

"Yet there seems to be a fundamental oversight in the guidelines produced: the omission of any reference to the needs and welfare of working animals, ..."

"Were working animals omitted from the guidelines..."

The original paper states:

"These guidelines apply to domestic animals commonly used in development projects in the Global South, such as cattle, goats, sheep, pigs, donkeys, camels, and poultry. Recipients may receive these animals to raise for food, sale, or breeding, or they may be used in scientific research. Whatever the context, international development organisations will ensure that animals have the “Five Freedoms” [25], which underpin animal welfare legislation worldwide [26]."

My view is that the key approach of the comment is not justified , specifically  stating there is  "fundamental oversight" is explicit, whereas the paper was not explicit, i.e. " Whatever the context,...".

This is more than semantics, it as such divides rather than moves to remind that working animals are included.  In short a more constructive tone in what is a valid critique would, in my view, be more appropriate.

Author Response

Comment: The submitted comment states:

"Yet there seems to be a fundamental oversight in the guidelines produced: the omission of any reference to the needs and welfare of working animals, ..."

"Were working animals omitted from the guidelines..."

The original paper states:

"These guidelines apply to domestic animals commonly used in development projects in the Global South, such as cattle, goats, sheep, pigs, donkeys, camels, and poultry. Recipients may receive these animals to raise for food, sale, or breeding, or they may be used in scientific research. Whatever the context, international development organisations will ensure that animals have the “Five Freedoms” [25], which underpin animal welfare legislation worldwide [26]."

My view is that the key approach of the comment is not justified , specifically  stating there is  "fundamental oversight" is explicit, whereas the paper was not explicit, i.e. " Whatever the context,...".

This is more than semantics, it as such divides rather than moves to remind that working animals are included.  In short a more constructive tone in what is a valid critique would, in my view, be more appropriate.

Response: We appreciate your feedback and by no means meant to divide concern, rather to remind the readers that working animals are often critical in IDO efforts and it would have been good to see this recognised in the Ssuna et al paper rather than them falling into the ‘whatever the context’ catch all. The editor has already requested the wording is changed to ‘significant oversight’ rather than ‘fundamental oversight’ as in our original manuscript and this has been changed. Further we have now amended the wording of the second sentence you picked up on so it now reads ‘were not specifically mentioned in the guidelines’. We do however, believe that working animals do warrant a specific mention in the context of these guidelines because they often support IDO activities as well as sometimes being part of their provision.

Reviewer 2 Report

Comments and Suggestions for Authors

As this is a comment paper I will keep my comments brief. This warrants discussion and is appropriate to raise concerns regarding the role of working animals. Specific mention of animal for manual labour (perhaps where greatest quantity of animals are, and where welfare risks are greatest beyond production animals in the Global South) would have been beneficial in the opening paragraph of section 3 for this clarity. However, it is important to note that the species identified in your comment are specifically identified in section 3.1. 

Author Response

Comment: As this is a comment paper I will keep my comments brief. This warrants discussion and is appropriate to raise concerns regarding the role of working animals. Specific mention of animal for manual labour (perhaps where greatest quantity of animals are, and where welfare risks are greatest beyond production animals in the Global South) would have been beneficial in the opening paragraph of section 3 for this clarity. However, it is important to note that the species identified in your comment are specifically identified in section 3.1.

Response: We acknowledge that donkeys and camels are mentioned in section  3.1 of the Ssuna et al paper. This section of the paper reads "These guidelines apply to domestic animals commonly used in development projects in the Global South, such as cattle, goats, sheep, pigs, donkeys, camels, and poultry. Recipients may receive these animals to raise for food, sale, or breeding, or they may be used in scientific research." This does not specifically mention working animals and neither does it recognise that working animals are used to deliver and support other species of animals provided by development projects.

Round 2

Reviewer 1 Report

Comments and Suggestions for Authors

Thank you for the changes